# Geography is a stronger predictor of diversification of monogenean parasites (Platyhelminthes) than host relatedness in characin fishes of Middle America

Fernando Alda[1,2☯*], Edgar F. Mendoza-Franco[3☯], William Hanson-Regan[4], Ruth G. Reina[2†], Eldredge Bermingham[2], Mark E. Torchin[2,5]

**1** Instituto de Investigación en Recursos Cinegéticos (IREC; CSIC-UCLM-JCCM), Ciudad Real, Spain, **2** Smithsonian Tropical Research Institute, Balboa, Ancón, Republic of Panama, **3** Instituto de Ecología, Pesquerías y Oceanografía del Golfo de México (EPOMEX), Universidad Autónoma de Campeche, San Francisco de Campeche, Campeche, Mexico, **4** Department of Biology, Geology and Environmental Science, University of Tennessee at Chattanooga, Chattanooga, Tennessee, Unites States of America, **5** Marine Science Institute, University of California, Santa Barbara, California, United States of America

† Deceased.
☯ These authors contributed equally to this work.
* fernando.alda@csic.es

## Abstract

Host-parasite associations have historically been considered compelling examples of coevolution and useful in examining cospeciation. However, modern molecular methods have revealed more complex dynamics than previously assumed, with host-switching events appearing commonly across taxa and challenging traditional views of strict coevolution in host-parasite relationships. Monogenean parasites are considered highly host-specific and have long served as models for probing evolution of host-parasite associations, particularly in differentiating geographic and phylogenetic patterns of parasite diversification. We investigated the phylogeographic patterns of monogenean ectoparasites associated with four species of characin fishes across Panama, Nicaragua, and Mexico. We hypothesize that parasite diversity and community structure are more strongly correlated with host species (suggesting cospeciation) than with geographic location (indicative of allopatric speciation). We found high genetic differentiation among parasites and their hosts across different locations. However, while geography explained the genetic structure of both host fishes and parasites, the observed patterns were neither congruent nor parallel. Parasite community structure and genetic similarity were consistently better explained by geographic location than by host species identity, although both factors played a significant role. Contrary to our predictions, we found no evidence of cospeciation. Instead, the diversification of these monogenean parasites appears to be primarily driven by their ability to switch hosts. At this taxonomical scale, host-switching is

**Data availability statement:** All resulting sequences have been deposited in the NCBI GenBank database under accession numbers PQ256093-256142 and PQ275828-275931 (S1 Table).

**Funding:** Funding for sequencing was provided by the Smithsonian Institution's DNA Barcode Network to F.A and M.E.T. E.F.M-F. received funding from a Smithsonian Tropical Research Institute postdoctoral fellowship. W.H.-R. was funded through a grant from the University of Tennessee at Chattanooga Center of Excellence in Applied Computational Science and Engineering (CEACSE) awarded to F.A. The funders had no role in study design, data collection and analysis, decision to publish, or preparation of the manuscript.

**Competing interests:** The authors have declared that no competing interests exist.

mediated by the geographical proximity of potential hosts, underscoring the importance of spatial factors in parasite evolution.

## Introduction

Parasites with high host specificity and direct transmission, such as monogenean ectoparasites of fishes, represent valuable models for phylogeographic comparative studies [1–3]. Monogeneans are often restricted to a single host species or closely related hosts, occupying specific microhabitats within the hosts [4]. Their direct transmission relies on short-lived free-swimming larval stages to infect new hosts, creating an intimate association that is expected to produce congruent phylogenetic patterns between hosts and parasites. While some studies support this expectation [2,3], others suggest that monogeneans can disperse effectively and frequently switch hosts, often without cospeciating [5,6]. These contrasting findings may reflect variations in environmental conditions, geographical scales, or taxonomic levels of the hosts and parasites being studied. For instance, cospeciation patterns may be more common at higher host taxonomic levels (e.g., genera or families), while host switching is more prevalent at species or population levels [7]. Thus, monogeneans provide an ideal system to test host-parasite coevolution and disentangle geographic and phylogenetic factors shaping parasite diversification [4,8,9].

Historically, parasite phylogenies were expected to mirror those of their hosts [10], and early studies often adhered to a "maximum cospeciation" framework, assuming cospeciation as the main mode of diversification. Incongruences were typically attributed to parasite extinction rather than host switching [11,12]. However, this paradigm has shifted. Recent studies demonstrate that congruent host-parasite phylogeographic patterns are rare [13] and that speciation driven by host switching is as common, if not more so, than cospeciation [14].

The success of host switching depends on three critical, interrelated factors: opportunity, compatibility, and conflict resolution [15,16]. Opportunity refers to the spatial and temporal proximity of the parasite to a potential new host, influenced by the geographic range, ecological overlap, and interactions among hosts. If opportunity exists, the parasite must navigate compatibility, overcoming the physical, physiological, and immunological barriers of the new host. Compatibility is often influenced by host relatedness and the parasite's level of specialization. Finally, conflict resolution involves the parasite establishing a stable, long-term relationship with the new host, requiring adaptations to its immune defenses and ecological niche [12,15–17].

Opportunity is often shaped by geographic and trophic distribution, while compatibility, and conflict resolution reflect inheritable traits of both parasites and hosts that influence their interaction [12,16]. By integrating these factors, host-parasite interactions can be broadly understood as driven by two primary forces: 1) geography and ecology, and 2) genetic and phenotypic similarity. This framework highlights the interplay between external environmental factors and intrinsic biological traits in shaping host-parasite coevolution.

In this study, we examine characin fish species (Acestrorhamphidae: Acestrorhamphinae and Stevardiidae: Landoninae *sensu* Melo et al. [18]) and their monogenean gill parasites (Platyhelminthes: Monogenea). Among the Neotropical characiforms, *Astyanax* Baird and Girard 1854 stands out as the most diverse, widespread, and abundant genus [19]. *Astyanax* species are distributed from Patagonia in Argentina to Texas in the United States [20]. However, their dispersal from South America (SA) into and through Middle America (MA) is relatively recent (~8 Ma) and constrained by freshwater connections [21,22].

The parasite communities of *Astyanax* are both highly diverse and often host-specific. For example, up to nine parasite species from six genera have been reported within a single *Astyanax* host [23,24]. Several monogenean genera are specialists of characins, while others exhibit a narrower host range and parasitize exclusively *Astyanax* species. These *Astyanax*-specialist parasites form a "core fauna" of at least 18 species in 8 genera that are widely distributed across species and drainage basins [6,23,25,26]. While the rapid diversification of *Astyanax* in MA suggests a low diversification rate for their monogenean parasites, the broad geographic distribution of these parasites raises questions about the processes shaping their communities. Molecular studies, particularly phylogeographic analyses, are essential to elucidate the roles of drainage vicariance and host-parasite dispersal in shaping these biogeographical patterns [26].

We examined the phylogeographic structure of *Astyanax* spp. across Middle America (MA), from Panama to southeastern Mexico, and their monogenean gill parasites. Riverine fishes are constrained by stream geomorphology and hydrology, while their parasites are further restricted by their hosts, which act as mobile "islands" of habitat [27]. This system provides an exceptional opportunity to investigate the relative roles of host specificity, geographic isolation, and ecological opportunity in shaping parasite assembly and diversification. We hypothesized that if host genetics is the principal driver of parasite distribution and diversification, we would detect significant codivergence between host and parasite phylogeographic structures and/or a positive correlation between host and parasite genetic relatedness. Conversely, if geography is the main determinant, we would expect parasite community similarity to correlate more strongly with geographic location than host identity. By testing these hypotheses, we aim to clarify the relative roles of cospeciation, host switching, and allopatric processes in the diversification of tropical monogenean parasites and their fish hosts.

## Materials and methods

### Sampling of fish and parasites

We sampled individuals from three species of *Astyanax*: *Astyanax aeneus* (Günther 1860), *A. orthodus* Eigenmann 1907, and *A. panamensis* (Günther 1864), along with the related characin, *Eretmobrycon gonzalezoi* (Román-Valencia 2002), across eight locations in Panama, two in Nicaragua, and three in Mexico. We found *A. aeneus* at ten sites, *A. panamensis* at three, and *A. orthodus* and *E. gonzalezoi* at a single site each (Fig 1A and Table 1). The sampling localities correspond to six areas of fish endemism, or ichthyological provinces: Tuira, Chagres, Chiriquí-Santa María, Bocas, Mosquitia-San Juan, and Grijalva-Usumacinta (Fig 1A and Table 1). These regions are defined by historical processes such as dispersal, speciation, and extinction, which have influenced the diversity and distribution of freshwater fish communities. As such, they provide valuable context for understanding the biogeographic history of the region [28,29]. For subsequent geographical analyses, these areas of endemism serve as the primary sampling units (see below).

We collected fish using an electrofishing unit or seine nets and euthanized them with tricaine methanesulfonate (MS-222). All procedures complied with the regulations set by the Panamanian Autoridad Nacional del Ambiente (collection permits: SE/APO-1-13, SC/AP-4-15, SE/A/30-2019) and approved by the Smithsonian Tropical Research Institute IACUC committee (proposal number: 2015-0115-2018, 2016-0304-2019). For the genetic analysis of the fish, we dissected the gill arches with filaments or muscle tissue, which were preserved at ambient temperature in a saturated salt solution containing 20% DMSO and 0.5 M EDTA (pH 8). Morphological voucher specimens for all individuals were deposited in the

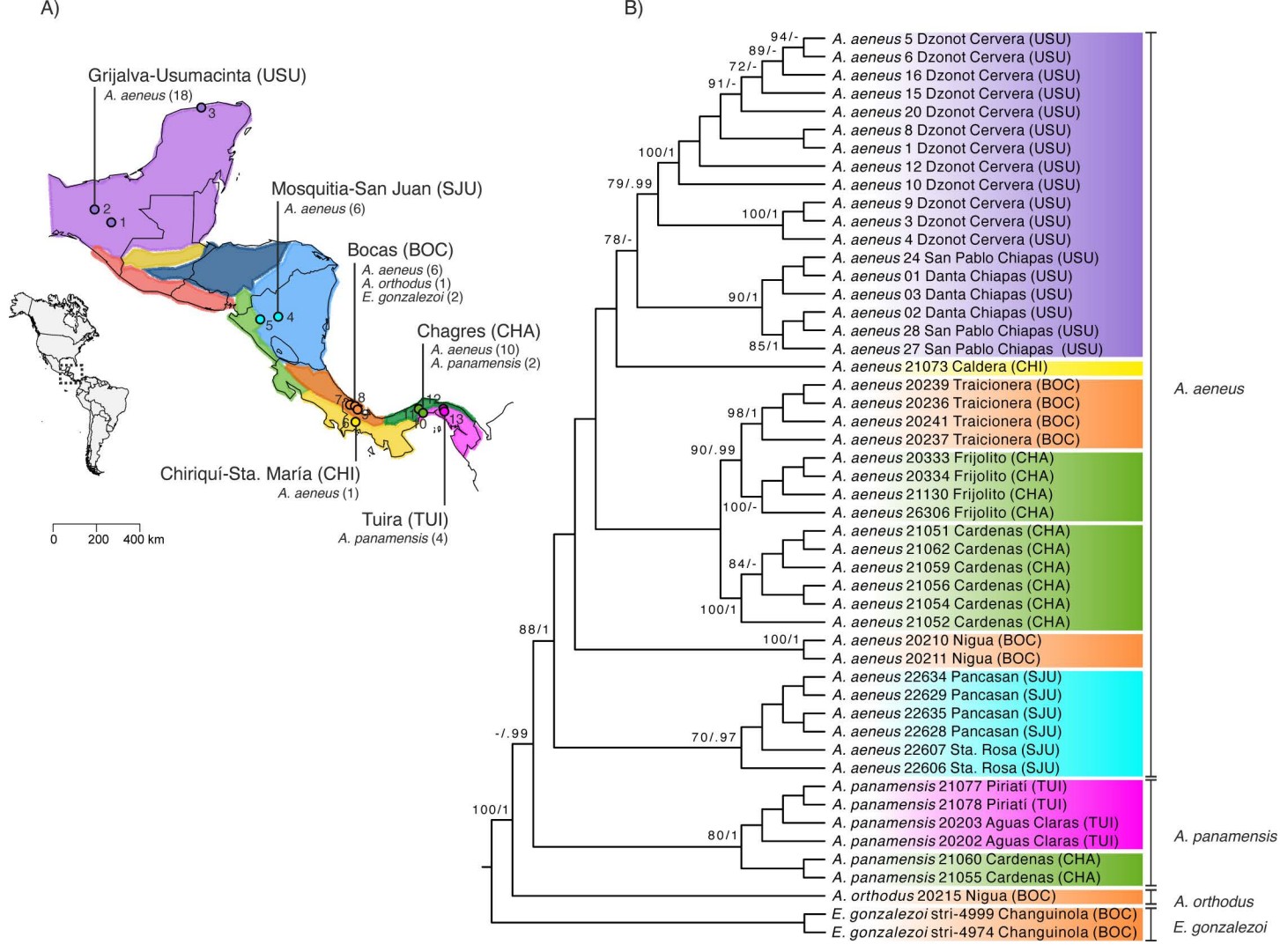

**Fig 1. Sampling and phylogenetic analysis of fish hosts. (A)** Map of Middle America showing localities and species of fish sampled. Numbers next to sampling points correspond to the localities in Table 1. Black lines indicate political borders, and colored regions indicate areas of endemism following Matamoros et al. [29]. The dashed-line square in the inset map indicates the area of study. **(B)** Cladogram inferred using mitochondrial *ATPase 8/6* gene sequences from all host fish sampled. The topology shown corresponds to the Maximum Likelihood (ML) tree and the values above branches indicate nodal support for the ML and the Bayesian Inference analyses (Bootstrap Support (BS)/ Bayesian Posterior Probability (PP)). For clarity, only BS ≥ 70 and PP ≥ 90 are shown.

Neotropical Fish Collection (NFC-STRI) at the Smithsonian Tropical Research Institute, Panama. For parasite analysis, gill arches with attached monogeneans were dissected immediately after fish collection, preserved in vials with 96% ethanol, and stored at 4 °C until parasite isolation and DNA extraction. We removed monogeneans from the fish gills, dissected the haptor (attachment organ) from each specimen and mounted it in Gray and Wess' medium for morphological identification. We retained the remaining of the body for DNA extraction. We examined the sclerotized parts of the attachment organ (i.e., anchors, hooks, and dorsal/ventral bars) under an Olympus microscope with a drawing attachment tube at 100X magnification for species determination [23].

**Table 1. List of fish host and parasite species isolated across 13 collecting sites in six areas of endemism in Middle America.**

| Area of endemism | Locality | Host species | No. Hosts | Parasite species | | | | | | |
|---|---|---|---|---|---|---|---|---|---|---|
| | | | | C. costaricensis | D. kabatai | D. orthodusus | P. heteroancistrium | Gyrodactylus sp. | U. strombicirrus | Dactylogyridae gen sp. |
| Grijalva-Usumacinta (USU) | 1. San Pablo Chiapas | A. aeneus | 3 | 1 | 3 | | 2 | | 3 | |
| | 2. Danta Chiapas | A. aeneus | 3 | 1 | | | 13 | | | |
| | 3. Dzonot Cervera | A. aeneus | 12 | 1 | 3 | | 15 | | | |
| Mosquitia-San Juan (SJU) | 4. Pancasápn | A. aeneus | 4 | 12 | 1 | | | | | |
| | 5. Sta. Rosa | A. aeneus | 2 | | 2 | | | | | |
| Chiriquí-Sta. María (CHI) | 6. Caldera | A. aeneus | 1 | | 1 | | | | | |
| Bocas (BOC) | 7. Bongie | E. gonzalezoi | 2 | 3 | | | | | 1 | |
| | 8. Nigua | A. aeneus | 2 | 2 | | | 4 | | | |
| | | A. orthodus | 1 | | | 2 | | | | |
| | 9. Traicionera | A. aeneus | 4 | 6 | 2 | | | | 2 | |
| Chagres (CHA) | 10. Cárdenas | A. panamensis | 2 | 4 | | | | | | |
| | | A. aeneus | 6 | 2 | 1 | | 2 | | 1 | |
| | 11. Frijolito | A. aeneus | 4 | 5 | 1 | | 1 | 1 | | |
| Tuira (TUI) | 12. Aguas Claras | A. panamensis | 2 | | 2 | | | | | |
| | 13. Piriatí | A. panamensis | 2 | 1 | | | 1 | 1 | 7 | 1 |
| | | | 50 | 38 | 16 | 2 | 39 | 1 | 7 | 1 |

## Extraction, amplification and sequencing of DNA

We extracted fish DNA from gill or muscle tissue using a standard phenol-chloroform method and amplified the entire mitochondrial *ATPase 8/6* genes using primers L8331 (5'-AAA GCR TYR GCC TTT TAA GC-3') and H9236 (5'-GTT AGT GGT CAK GGG CTT GGR TC-3') [30]. We carried PCR in a final volume of 20 µl including 1 U of AmpliTaq DNA polymerase (Applied Biosystems), 1X reaction buffer (5 mM KCl, 1 mM Tris–HCl), 2.25 mM $MgCl_2$, 0.5 µM of each primer, 0.2 mM of each dNTP, and 1–3 µl of DNA extract. Thermocycler conditions for *ATPase 8/6* consisted of: initial denaturation at 94 °C for 4 min, denaturation at 93 °C for 45 sec, annealing at 55 °C for 45 sec, and extension at 72 °C for 45 sec, for a total of 35 cycles, and a final elongation at 72 °C for 10 min. We purified PCR products by manually excising them from 2% low-melting-point agarose gels and incubating them at 45 °C for 3 h with GELase (Epicentre).

We extracted DNA from parasites using the DNeasy Blood & Tissue kit (Qiagen) and eluting DNA in 75 µl of AE buffer. We amplified the 5' region of the mitochondrial cytochrome c oxidase subunit I *(COI)* gene using primers COI-F trema (5'-TTT TTT GGG CAT CCT GAG GTT TAT-3') (originally JB3 in [31]) and COI-R trema (5'-CAA CAA ATC ATG ATG CAA AAG G-3') [32]. We performed PCR in a final volume of 25 µl containing 0.5 U of AmpliTaq DNA polymerase (Applied Biosystems), 1X reaction buffer (5 mM KCl, 1 mM Tris–HCl), 2.0 mM $MgCl_2$, 0.5 µM of each primer, 0.2 mM of each dNTP, and 1–5 µl of DNA extract. Thermocycling conditions were as follows: initial denaturation at 94 °C for 4 min, followed by 35 cycles of 1 min at 94 °C, 1 min at 40 °C, and 1 min 30 sec at 72 °C, and 10 min of final elongation at 72 °C. We separated and extracted PCR products as described above.

We sequenced the purified PCR products from both fish and parasites in an ABI3130*xl* automated sequencer using the BigDye terminator v.3.1 kit (Applied Biosystems) and the same primers used for amplification. All resulting sequences have been deposited in the NCBI GenBank database under accession numbers PQ256093-256142 and PQ275828-275931 (S1 Table).

## Phylogenetic inference

We inferred phylogenetic trees for the fish and parasite mitochondrial sequence data using Maximum Likelihood (ML) and Bayesian Inference (BI). We used jModelTest v.2.1.1. [33] and the corrected Akaike information criterion (AICc) to select the best fit model of nucleotide substitution. In the fish dataset, we considered the *ATPase 8* and *ATPase 6* genes as separate partitions, and in the parasite dataset, we considered the *COI* alignment as a single partition. We performed 20 ML searches to find the best ML tree in RAxML v.7.2.6 [34] and assessed clade support using 500 bootstrap replicates. To construct the BI trees, we conducted four independent MCMC runs (with four chains each) for $10 \times 10^6$ generations, sampling every 1000 generations, and discarding the first 25% of samples as burn-in. We assessed convergence and sampling independence between and within MCMC runs via visual inspection of ln *L* scores vs. generation time in Tracer v.1.7 [35], Estimated Sample Size (ESS > 200), and Potential Scale Reduction Factor (PSRF < 0.01), as implemented in MrBayes. We rooted the fish tree with the sequences of *E. gonzalezoi* and the parasite tree with *Gyrodactylus* sp.

## Cophylogenetic analysis

We built a cophylogenetic plot of ML trees for fish hosts and their parasites ML using the *cophylo* function in the R package phytools v.0.7-20 [36]. To assess congruence between host and parasite phylogenies, we employed two complementary approaches: an event-based method and a distance-based method.

**Event-based method.** We used the software Jane v.4.01 [37] to reconcile host and parasite tree topologies by minimizing the overall cost of coevolutionary events based on a predefined cost matrix. The five evolutionary events considered were: Cospeciation, in which parasite speciation occurs together or following host speciation; duplication, in which the parasite speciates within a single host species; loss or lineage sorting, in which one of the parasite lineages is lost after host speciation; failure to diverge, in which the parasite fails to speciate and persists in both hosts after

speciation; and host switching, where the parasite is able to colonize and speciate in a new and unrelated host. To explore different evolutionary scenarios, we ran Jane v.4.0 under nine cost regimes. For example, we penalized cospeciation events to simulate low parasite host-specificity, penalized host switches to simulate vicariant host speciation, or assigned equal weights to all events (Table 2). We set the population size to 100, and the number of generations to 1000 to evaluate the significance of coevolutionary events. Statistical significance was assessed by comparing the observed total cost for each regime to a null distribution generated from 500 random parasite trees. If the observed costs were significantly lower than the null distribution ($p < 0.05$), we inferred a global signal of cospeciation for that cost regime. We included unresolved nodes as polytomies because Jane 4.0 can resolve soft polytomies in both trees to minimize the cost of the co-phylogenetic reconstruction.

**Distance-based method.** We used the program ParaFit [38] to test the global congruence of host and parasite phylogenies. ParaFit uses matrices of patristic distances (summed branch lengths along a phylogenetic tree) to evaluate the null hypothesis of random host-parasite associations. We tested the statistical significance of the global congruence and individual host-parasite links using 999 permutations of the association matrix. Unlike event-based methods, ParaFit does not identify specific coevolutionary events but accounts for cases where multiple parasites are associated with a single host—an advantage in our study, as this pattern was prevalent.

### Geographical analysis

As an alternative to host-parasite coevolution, we hypothesized that parasite diversification is driven by geography rather than host relatedness. To test whether parasite diversity and community structure are shaped by geographic location and/or host identity, we conducted a permutational multivariate analysis of variance (PERMANOVA) using the *adonis2* function in the R package *vegan* v.2.6–4 [39]. This method evaluates ecological and geographical determinants of parasite diversification [40,41]. We grouped hosts by species and geographic location, (i.e., areas of endemism *sensu* Matamoros et al. [29] and tested their influence on parasite community composition using Jaccard pairwise differences [42]. Additionally, we used PERMANOVA to evaluate whether host species or geographic location predicted genetic similarity among individual parasites, using a genetic distance matrix calculated in PAUP v.4.0a123 [43] based on the best-fit nucleotide substitution model estimated by jModeltest.

To assess the relative contribution of geographic location and host species in explaining parasite community composition and genetic differentiation, we first conducted a full PERMANOVA model including both predictors, followed by separate models for each factor independently. We then compared their $R^2$ values to evaluate their relative explanatory power. We used variance partitioning (*varpart* in vegan) to quantify the unique and shared variation explained by each predictor.

We performed pairwise post-hoc comparisons using *pairwise.adonis* in the pairwiseAdonis package [44] to assess differences in parasite community composition and genetic variation between levels of each categorical variable. The significance of each pairwise comparison was adjusted for multiple testing using the Benjamini-Hochberg false discovery rate correction.

Finally, we tested whether host genetic similarity or geographic distance between sampling locations predicted parasite species co-occurrence. We ran two logistic regressions (generalized linear model with logit link function) in R, using either a pairwise genetic distance matrix of fish hosts (calculated in PAUP v.4.0a123) or a matrix of pairwise linear geographical distances (in kilometers) as the explanatory variables. The response variable in both models was a binary matrix coding whether two hosts shared at least one parasite species (1) or not (0).

## Results

### Phylogeographic analysis

We sequenced the ATPase *8/6* mitochondrial genes from 50 characin fishes, including three species of *Astyanax* (*A. aeneus*, *A. panamensis*, and *A. orthodus*) and *E. gonzalezoi* (Table 1). The alignment spanned 853 bp

(ATPase 8: 168 bp and ATPase 6: 685 bp) and contained 394 polymorphic sites. jModelTest identified the best nucleotide substitution models as HKY + I for ATPase 8 and TIM1 + G for ATPase 6. The ML and BI phylogenetic trees were congruent for the major relationships, recovering three main lineages with strong support (bootstrap support, BS ≥ 80; posterior probability; PP ≥ 0.99). *Astyanax orthodus* formed the most divergent lineage, sister to the lineage comprising reciprocally monophyletic *A. panamensis* and *A. aeneus* (Fig 1B). All species and areas of endemism were genetically differentiated and formed independent clades. Based on this pronounced phylogeographic structure, we categorized hosts in the cophylogenetic analyses by both species taxonomy (four hosts) and geographically defined clades (11 hosts).

We isolated 104 individual monogenean parasites from fish gills and identified seven monogenean taxa through morphological analysis: *Characithecium costaricensis* (Price and Bussing 1967) Mendoza-Franco, Reina and Torchin 2009 (n = 38); *Diaphorocleidus kabatai* (Molnar, Hanek and Fernando 1974) Jogunoori, Kritsky and Venkatanarasaiah 2004 (n = 16); *Diaphorocleidus orthodusus* Mendoza-Franco, Reina and Torchin 2009 (n = 2); *Gyrodactylus* sp. (n = 1), *Palombitrema heteroancistrium* (Price and Bussing 1968) Suriano 1997 (n = 39); *Urocleidoides strombicirrus* (Price and Bussing 1967) Kritsky and Thatcher 1974 (n = 7, incertae sedis in *Urocleidoides*), and Dactylogyridae gen sp. (n = 1). The most abundant parasites infecting *A. aeneus* were *P. heteroancistrium* and *C. costaricensis*. *Palombitrema heteroancistrium* was prevalent in the Grijalva-Usumacinta area but absent from San Juan and Chiriquí-Sta. María. *Characithecium costaricensis* was widespread in the southern areas of endemism but absent from Chiriquí-Sta. María, where only one fish was sampled. *Diaphorocleidus kabatai* was the only species present in all areas of endemism, infecting both *A. aeneus* and *A. panamensis*. In contrast, *D. orthodusus* was restricted to *A. orthodus* in the Bocas region of Panama.

The parasite *COI* dataset comprised 104 sequences, 779 bp long, with 463 polymorphic sites. jModelTest identified TIM3 + I + G as the best nucleotide substitution model the dataset. The ML and BI phylogenetic trees revealed three major parasite lineages (BS ≥ 83, PP = 1.0) with minor differences involving low supported relationships within them. Lineage I included *C. costaricensis* samples from the Mosquitia-San Juan area (Nicaragua), Grijalva-Usumacinta (Mexico), and Chagres and Chiriquí-Sta. María (Panama). Lineage II comprised *P. heteroancistrium* from Grijalva-Usumacinta (Mexico), and Tuira, Bocas, and Chagres (Panama); *D. orthodusus* from Bocas; *U. strombicirrus* from Grijalva-Usumacinta, Bocas and Chagres; and *D. kabatai* from San Juan, and all the Panama provinces (Bocas, Chiriquí-Sta. María, Chagres, and Tuira). In the ML tree, Dactylogyridae gen sp. from Tuira was included in this lineage with strong support (BS = 100), but its position was unresolved in the BI tree. Lineage III was sister to the first two and consisted of *C. costaricensis* infecting *E. gonzalezoi* from Bocas (Panama) (Fig 2A).

All parasite species except *C. costaricensis* and *D. kabatai* were recovered as monophyletic. Parasite populations showed high phylogeographic structure, with genetic differentiation aligning closely with sampling sites and areas of endemism, though geographically proximate areas did not always group as closest relatives (Fig 2A).

## Cophylogenetic analysis

The event-based cophylogenetic analysis in Jane revealed that host switches and lineage losses played significant roles in the speciation of monogenean parasites infecting *Astyanax* (Table 2). Reconstructions under nine cost regimes suggested statistically significant global costs (*p* < 0.05) for seven regimes when considering 11 host geographical clades. Regime 4, which did not penalize host switches and duplications and equally penalized cospeciation, losses, and failures to diverge, minimized costs. This model inferred 0 cospeciation events, 6 duplications, 18 host switches, 15 losses, and 9 failures to diverge. In contrast, the highest-cost regime penalized host switches, losses, and failures to diverge while allowing cospeciation at no cost, further highlighting the importance of non-cospeciation events in monogenean parasite diversification. When the analysis used only four host species (species taxonomy), Regime 4 again minimized costs, but none of the models were statistically significant.

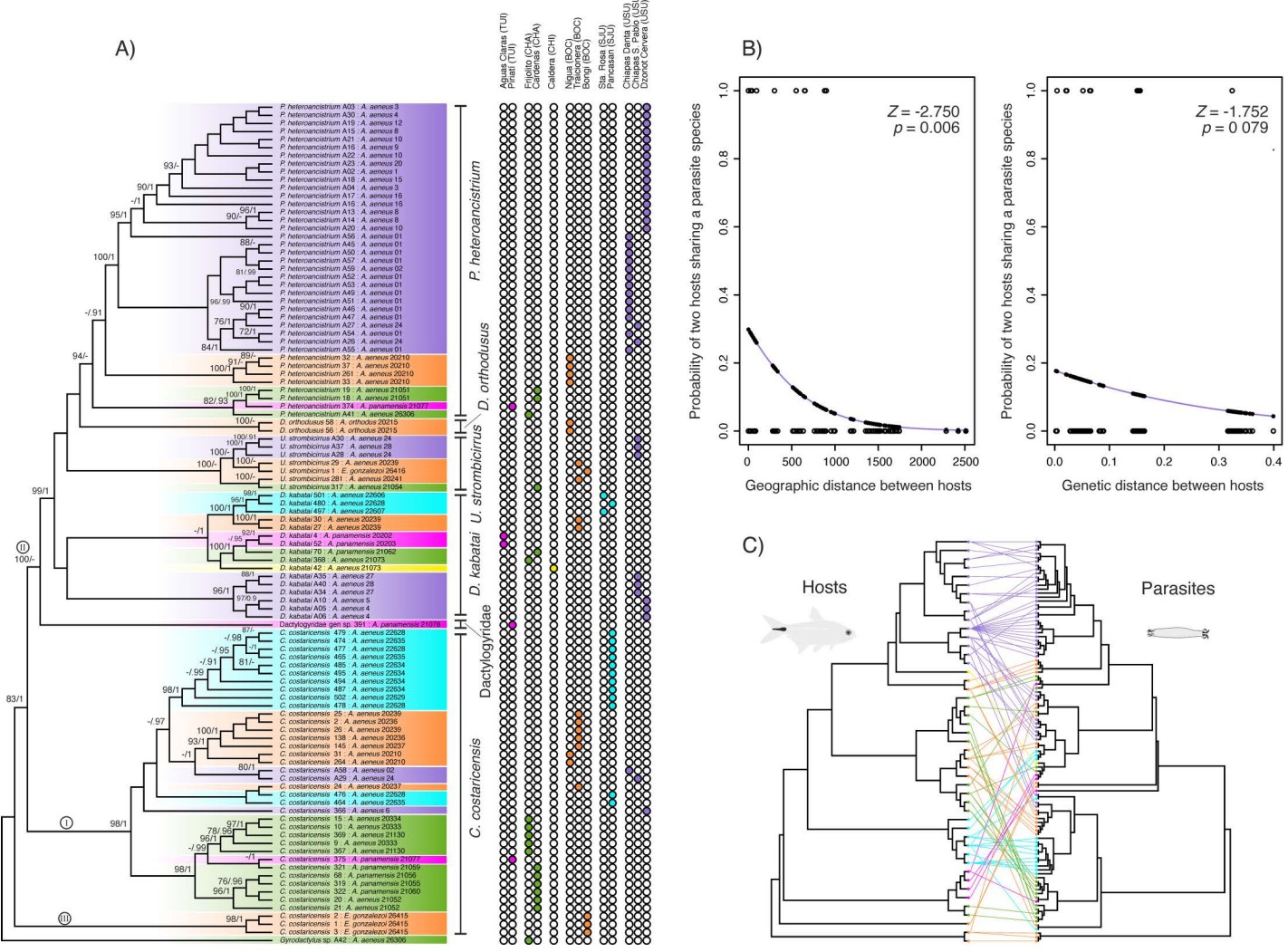

**Fig 2. Phylogenetic analysis of monogenean parasites and relationship with their fish hosts. (A)** Maximum Likelihood cladogram inferred using mitochondrial *COI* gene sequences of monogenean parasites infecting characin fishes. Values above branches indicate nodal support (BS/PP). Only BS ≥ 70 and PP ≥ 90 are shown. Roman numerals show lineages mentioned in the text. Tip labels indicate parasite species and sample code and the species and code of its host separated by a colon. Colored bullets represent the sampling locality of each sequence. **(B)** Logistic regression plots for the probability of two hosts sharing the same parasite species depending on the geographical and genetic distance between them. In blue is the estimated logistic regression line. Empty dots are the actual responses, and black dots are the predicted probabilities (fitted values). **(C)** Cophylogenetic plot showing the associations between characin fish hosts (left) and their monogenean parasites (right). Phylogenetic trees are ML trees in Figs 1B and 2A. Line colors correspond to areas of endemism. (Full figure with sample names available in S1 Fig).

The distance-based coevolution tests in Parafit found no significant global congruence between host and parasite phylogenies, either across the four host species (ParaFitGlobal = 0.0248, p = 1) or the 11 geographical clades (ParaFitGlobal = 0.0265, p = 0.929), indicating that the null hypothesis of each parasite species being randomly associated with hosts along the fish phylogenetic tree could not be rejected. Individual host-parasite links were also non-significant, except for two associations: *P. heteroancistrium* and *C. costaricensis* exclusively infecting *A. aeneus* (Grijalva-Usumacinta in Mexico), and *E. gonzalezoi*, respectively (Figs 2C and S1).

**Table 2. Output of cophylogenetic analysis calculated using 9 regimes with different cost schemes and considering four host species and 11 host geographic clades.**

| | 4 Hosts | | | | | | | |
| --- | --- | --- | --- | --- | --- | --- | --- | --- |
| | Cost regime | C | D | HS | L | FD | Total cost | *p*-value (RPT) |
| Regime 1 | 01111 | 1 | 8 | 15 | 8 | 8 | 39 | 0.297 |
| Regime 2 | 01222 | 1 | 18 | 5 | 9 | 8 | 62 | 0.373 |
| Regime 3 | 11111 | 1 | 8 | 15 | 8 | 8 | 40 | 0.263 |
| Regime 4 | 10011 | 0 | 8 | 16 | 9 | 8 | 17 | 0.244 |
| Regime 5 | 21110 | 0 | 8 | 16 | 9 | 8 | 33 | 0.260 |
| Regime 6 | 21111 | 0 | 8 | 16 | 9 | 8 | 41 | 0.254 |
| Regime 7 | 21100 | 0 | 8 | 16 | 9 | 8 | 24 | 1.000 |
| Regime 8 | 01121 | 1 | 8 | 15 | 8 | 8 | 47 | 0.260 |
| Regime 9 | 01211 | 1 | 18 | 5 | 9 | 8 | 45 | 0.491 |
| | 11 Hosts | | | | | | | |
| | Cost regime | C | D | HS | L | FD | Total cost | *p*-value (RPT) |
| Regime 1 | 01111 | 2 | 1 | 21 | 15 | 9 | 46 | 0.026* |
| Regime 2 | 01222 | 1 | 6 | 17 | 15 | 9 | 88 | 0.027* |
| Regime 3 | 11111 | 0 | 6 | 18 | 15 | 9 | 48 | 0.004** |
| **Regime 4** | **10011** | **0** | **6** | **18** | **15** | **9** | **24** | **0.002**** |
| Regime 5 | 21110 | 0 | 6 | 18 | 15 | 9 | 39 | 0.005** |
| Regime 6 | 21111 | 0 | 6 | 18 | 15 | 9 | 48 | 0.003** |
| Regime 7 | 21100 | 0 | 6 | 18 | 35 | 9 | 24 | 1.000 |
| Regime 8 | 01121 | 2 | 1 | 21 | 15 | 9 | 61 | 0.0013** |
| Regime 9 | 01211 | 3 | 6 | 15 | 17 | 9 | 62 | 0.097 |

Abbreviations correspond to the evolutionary events described in the text: cospeciation (C); duplication (D), host-switch (HS), loss or lineage sorting (L), failure to diverge (FD).

The total costs, number of individual evolutionary events, and *p*-values based on 500 random parasite trees (RPT) are shown for each regime.

The regime with the lowest significant total cost is highlighted in bold.

* *p* < 0.05, ** *p* < 0.01.

## Community and geographical analysis

PERMANOVA tests indicated that both geography (area of endemism) and host species significantly explained parasite community composition. Geography had a stronger effect ($F = 2.95$, df $= 5$, $R^2 = 0.261$, $p = 0.003$) compared to host species ($F = 2.294$, df $= 2$, $R^2 = 0.081$, $p = 0.025$). This pattern was further supported when assessing the explanatory variables together and separately, where the effect of host species alone was not significant (S2a Table). Variance partitioning analysis revealed that site alone explained 18.05% of the variation after accounting for host species identity, whereas host species alone accounted for 5.16% of the variation after controlling for site. A large proportion of the variation (77.77%) remained unexplained.

For parasite genetic similarity, geography was again the stronger predictor ($F = 9.347$, df $= 5$, $R^2 = 0.307$, $p = 0.001$) compared to host species ($F = 2.692$, df $= 3$, $R^2 = 0.053$, $p = 0.003$). However, both factors were highly significant ($p < 0.01$) when considered together and separately (S3a Table). The unique contribution of geographic location to variance explained, after controlling for host species, was 28.25%, while host species alone accounted for 3.51%. A small fraction (0.5%) was attributed to the combined effect of both factors, and 67.69% of the variance remained unexplained.

Differences between host species were primarily driven by variation in parasite community composition in *E. gon-zalezoi*. Pairwise comparisons between sampling localities consistently identified significant differences involving the

Grijalva-Usumacinta region, specifically Grijalva-Usumacinta vs. Bocas, and Grijalva-Usumacinta vs. Tuira (S2b Table). In contrast, parasite genetic similarity was not significantly associated with host species identity, whereas all pairwise comparisons of geographic regions involving Grijalva-Usumacinta and Mosquitia-San Juan (except for Mosquitia-San Juan vs. Chiriquí-Sta. María) were significant (S3b Table).

Logistic regression showed a significant negative relationship between geographic distance and the likelihood of hosts sharing the same parasite species (estimate = -0.002, SE = 0.001, Z = -2.750, $p$ = 0.006). Host genetic distance was a marginally non-significant predictor (estimate = -5.318, SE = 3.036, Z = -1.752, $p$ = 0.079) (Fig 2B). The interaction between geographic and genetic distances was non-significant ($p$ = 0.799), and the simpler model excluding the interaction had a better fit (AIC = 66.775 vs. 68.706).

## Discussion

This study represents the first comparative analysis of host-parasite phylogeographic patterns across Middle America (MA), adding to the extensive body of research on helminths parasitizing Neotropical freshwater fishes. We observed significant phylogeographic structure in both *Astyanax* species and their monogenean parasites, but the lack of congruence between their patterns suggests that different factors or responses are driving their diversification.

The gill monogeneans identified in this study are consistent with the previously documented "core-fauna" parasitizing characins including *Astyanax* spp. in MA [26]. Their geographical distributions align with earlier findings, with *C. costaricensis*, *P. heteroancistrium,* and *D. kabatai* exhibiting the widest ranges—from the Grijalva-Usumacinta area in Mexico to the Tuira basin in Panama—and the highest prevalence, particularly in northern regions. In local communities within southern Mexico, *C. costaricensis* and *P. heteroancistrium* were dominant species [6]. Interestingly, sites with a high prevalence of *C. costaricensis* typically exhibited a low abundance or absence of *P. heteroancistrium*, and vice versa. For example, in Pancasán (Mosquitia-San Juan, Nicaragua), *C. costaricensis* was dominant, while *P. heteroancistrium* was absent. Conversely, in Danta Chiapas and Dzonot Cervera (Grijalva-Usumacinta, Mexico), *P. heteroancistrium* was most abundant, with only one individual of *C. costaricensis* found at each site. These contrasting patterns may reflect negative interactions between the two species [6]. However, this relationship was not evident in southern areas of higher biodiversity, where species evenness is greater, potentially reducing the influence of density-dependent interactions.

One notable absence in our findings was *Anacanthocotyle anacanthocotyle* Kritsky and Fritts 1970, a species often dominant in *Astyanax* monogenean parasite communities and considered part of the characin "core-fauna" [6,26]. This species, typically associated with external surfaces like skin, fins, or anus [6,45–47], may have been missed because our sampling focused on gill parasites. Although *A. anacanthocotyle* has been reported in the gills of SA characins [48], its absence in our study could also result from its endemicity, low abundance, rarity, or sampling stochasticity. Similarly, we found only one specimen of *Gyrodactylus* sp., a parasite typically found attached to the fish fins but not considered part of the characin "core-fauna" due to its broader host range [26,46,49]. Its scarcity in our samples could be attributed to its attachment preference for external surfaces or sampling limitations.

This study is one of the few to co-infer phylogenetic relationships between Neotropical monogenean parasites and their hosts [3,50,51]. The mitochondrial *COI* phylogenetic analyses supported the monophyly of most monogenean species while revealing possible instances of cryptic diversity. For example, *C. costaricensis* was distinctly differentiated from other dactylogyrid genera but was not monophyletic due to a non-sister divergent lineage parasitizing exclusively *E. gonzalezoi* (Fig 2A). This pattern further suggests hidden diversity within *C. costaricensis*, particularly in hosts with restricted distributions.

In the dactylogyrid parasites, we observed a weakly supported sister relationship between two clades of *D. kabatai* from the Grijalva-Usumacinta and the lower MA areas. However, neither clade was closely related to the congeneric *D. orthodusus*. Similarly, the phylogenetic position of *U. strombicirrus* relative to other genera remained unresolved in all analyses. Despite its widespread occurrence in *Astyanax* populations in Colombia, Nicaragua, and Panama [23,52–54], no formal reassignment of *U. strombicirrus* from its current *incertae sedis* status has occurred. Previous studies proposed varying taxonomic placements

for *U. strombicirrus*. Rossin and Timi [53] suggested transferring it to *Characithecium*, while Zago et al. [55] proposed its inclusion in *Urocleidoides* Mizelle and Price 1964 based on its close relationship with species parasitizing gymnotiform hosts. Our results, however, indicate that *U. strombicirrus* found in *A. aeneus* and *E. gonzalezoi* form a monophyletic group closely related to species of *Diaphorocleidus*, such as *D. orthodusus* from *A. orthodus*, and *D. kabatai* suggesting a potential reassignment to *Diaphorocleidus*. Recent studies have also found *Diaphorocleidus* infecting *Astyanax lacustris* from Brazil nested within a clade that includes *U. strombicirrus* and other species parasitizing gymnotiform hosts [56].

Currently, *Urocleidoides* comprises 57 species parasitizing diverse hosts across Characiformes, Gymnotiformes, Cyprinodontiformes, and Siluriformes, while *Diaphorocleidus* includes 10 species parasitizing Characiformes [54–56]. However, our phylogenetic analysis underscores unresolved questions about the relationships within these lineages, suggesting the need for taxonomic revisions. This aligns with previous findings that emphasize inconsistencies in the current taxonomy [23,56]. Importantly, our use of the *COI* fragment as a molecular marker, while effective for species identification via DNA barcoding, may lack the resolution required for deeper evolutionary inferences [57].

Despite these limitations, our molecular data revealed a clear phylogeographic structure within parasite species, with areas of endemism generally corresponding to monophyletic groups (Fig 2A). Notably, the Chagres and Tuira areas showed the closest genetic affinities and shared lineages of *D. kabatai*, *P. heteroancistrium,* and *C. costaricensis*. Additionally, the Chagres and Bocas regions shared a lineage of *U. strombicirrus*, underscoring the transitional role of the Chagres basin between SA and MA faunas [29]. These geographic patterns were further supported by the PERMANOVA analyses, which revealed significant differences in both parasite genetic similarities and community composition between Upper and Lower MA regions (S2 and S3 Tables).

The mitochondrial *ATPase 8/6* gene tree of the fish hosts showed a geographic structure, with all areas or collection sites forming monophyletic groups within each *Astyanax* species (Fig 1B). While some host-parasite phylogeographic patterns appeared correlated—such as *A. aeneus* and *P. heteroancistrium* in Grijalva-Usumacinta (purple in Fig 2C), *A. aeneus* and *C. costaricensis* from Mosquitia-San Juan (blue in Fig 2C), or *E. gonzalezoi* and *C. costaricensis* from Bocas (orange in Fig 2C)—these associations likely reflect the dominance of single parasite clades in local communities and host endemicity rather than significant co-diversification. Statistical tests did not detect significant correlations between host and parasite phylogeographic patterns or genetic distances.

In contrast to monogeneans, the diversity and evolutionary history of *Astyanax* in MA are better understood. Ornelas-García et al. [22] identified at least six deeply divergent *Astyanax* lineages in MA. Our study corroborates this diversity, identifying widely distributed lineages (*A. aeneus* populations in Grijalva-Usumacinta and Mosquitia-San Juan) and restricted lineages (*A. aeneus* in Chiriquí-Sta. María, *A. panamensis* in Chagres, and *A. orthodus* in Bocas). Newly identified monophyletic groups, such as *A. aeneus* from Bocas and Chagres and *A. panamensis* from Tuira, further emphasize the greater diversity of lower MA driven by drainage isolation that resulted from the geological fragmentation and increased tectonic activity during the formation of the Isthmus of Panama [58–62].

Our findings suggest that *Astyanax* introduced parasites during dispersal from SA, losing some and acquiring others, creating a mosaic of parasite assemblages across MA [12,26]. All the monogenean genera that we found parasitizing *Astyanax* in Panama to Mexico are also known from SA, except *Diaphorocleidus*, which is endemic to MA [23]. These results align with the biogeographic core fauna hypothesis, which posits strong Neotropical affinities and pronounced host lineage specificity in MA parasite communities. Collectively, our evidence supports the close evolutionary ties between MA fish monogeneans and their South American origins, shaped by dispersal and in situ diversification [26].

These results highlight the pivotal role of geographical factors in driving the diversification and assemblage of monogeneans parasitizing *Astyanax* in MA. First, geography influences host dispersal and promotes isolation through geological history, shaping freshwater fish assemblages across geological blocks and areas of endemism, as well as through the current spatial separation of populations within basins [61,62]. Second, geographical proximity facilitates host-switching, further supported by the ecological plasticity and widespread distribution of *Astyanax* species and the immense diversity of characins [19,63,64].

Interestingly, monogeneans from the same geographic region were more genetically similar to one another than those parasitizing the same host species (S3a Table). This observation contrasts with studies showing higher phylogenetic proximity among congeneric monogenean species parasitizing the same or closely related hosts [50,65,66]. The discrepancy may arise from differences in evolutionary scale: studies investigating relationships at or above the family level often detect signals of cophylogeny, while studies focusing on closely related genera, like ours, reveal the phylogenetic limits of such relationships [7,25]. Despite this, species identity remained a significant predictor of parasite genetic similarity, consistent with hierarchical interactions between geography, host identity, and parasite assemblages, as observed in other Neotropical host-parasite systems [25].

Patterns of parasite community composition (beta diversity) further support this conclusion. Parasite communities differed significantly among areas of endemism and to a lesser extent among host species (S2 Table). Notably, genetic distances explained these differences more effectively than ecological measures such as Jaccard pairwise differences—based on taxonomic identity—as molecular data incorporate phylogenetic diversity and within-species geographic structure, providing a more comprehensive view of parasite assemblage variation.

Host genetic divergence also imposes eco-physiological constraints on parasites [16]. However, the shallow diversification of *Astyanax* species likely diminishes this effect, contributing to the observed low host specificity and endemicity of their monogenean fauna [25,26]. Coevolutionary models favoring host-switching over other processes (Table 2) support this hypothesis, aligning with evidence that host-switching is widespread and a major driver of monogenean speciation [7,50,67–69]. Moreover, host-switching is often influenced by geographic factors, as potential hosts living in sympatry are more likely to facilitate such events [14,50,70]. In this system, genetically similar hosts do not necessarily share the same monogeneans species. Instead, hosts in close geographic proximity are more likely to share similar parasite assemblages (Fig 2B). This finding underscores the influence of river basin configuration—both historical and contemporary—in shaping parasite distribution and occurrence. The sharper phylogeographic structure of hosts suggests that parasites capable of host-switching can overcome dispersal limitations, enabling them to attain broader distributions.

## Conclusions

Despite the apparent generalism of widespread parasites, the phylogeographic structure of monogeneans in MA reveals that local genetic variants often result from host-switching. This process has been interpreted as a form of adaptive radiation in generalist monogeneans [50,71]. Thus, the capacity to switch hosts likely drives parasite diversification, and such events are strongly influenced by geographical proximity. Overall, our phylogeographic analysis elucidates the complex diversification history of monogenean parasites in MA, shaped by the interplay of geographical isolation, host-switching, and local adaptation.

## Supporting information

**S1 Table. Table of sampling collection sites, fish samples collected, and monogenean gill-parasites.** Coordinates are provided for each sample, and GenBank accession numbers. Sequence names as used in Figs 1 and 2.
(XLSX)

**S2 Table. Output of PERMANOVA analyses.** (a) using Jaccard distance of parasite community composition as the dependent variable and host species (species identity) and sampling site (area of endemism) as explanatory variables. Outputs of the full model (both explanatory variables together) and separate models for each explanatory variable are shown. (b) Output of pairwise PERMANOVA analyses using Jaccard distance of parasite community composition as the dependent variable and all the pairs of explanatory variables (host species and area of endemism).
(XLSX)

**S3 Table. Output of PERMANOVA analyses.** (a) using parasite genetic distance (*COI* mitochondrial gene) as the dependent variable and host species (species identity) and sampling site (area of endemism) as explanatory variables. Outputs of the full model (both explanatory variables together) and separate models for each explanatory variable are shown. (b) Output of pairwise PERMANOVA analyses using genetic distances (COI mitochondrial gene) of parasites as the dependent variable and all the pairs of explanatory variables (host species and area of endemism).
(XLSX)

**S1 Fig. Cophylogenetic plot showing associations between characin fish hosts and their monogenean parasites.** Phylogenetic trees are ML trees in Figs 1B and 2A. Line colors correspond to areas of endemism.
(PDF)

## Acknowledgments

The authors are grateful to E. Ríos Montenegro, A. Castillo, A. Terrero, C. Schlöder, E. Thompson, G. Arjona-Torres, and F. Puc-Itza for their assistance in the field and laboratory, and to Rigoberto González for curatorial assistance at STRI's Neotropical Fish Collection.

## Author contributions

**Conceptualization:** Fernando Alda, Edgar F. Mendoza-Franco, Mark E. Torchin.

**Data curation:** Fernando Alda, Edgar F. Mendoza-Franco, Ruth G. Reina.

**Formal analysis:** Fernando Alda, William Hanson-Regan.

**Funding acquisition:** Fernando Alda, Eldredge Bermingham, Mark E. Torchin.

**Investigation:** Fernando Alda, Edgar F. Mendoza-Franco, William Hanson-Regan.

**Methodology:** Fernando Alda, Edgar F. Mendoza-Franco, William Hanson-Regan, Ruth G. Reina.

**Project administration:** Fernando Alda, Mark E.Torchin.

**Resources:** Eldredge Bermingham, Mark E. Torchin.

**Supervision:** Fernando Alda, Eldredge Bermingham, Mark E. Torchin.

**Validation:** Fernando Alda, Edgar F. Mendoza-Franco.

**Visualization:** Fernando Alda.

**Writing – original draft:** Fernando Alda, Edgar F. Mendoza-Franco, William Hanson-Regan.

**Writing – review & editing:** William Hanson-Regan, Eldredge Bermingham, Mark E.Torchin.

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
