## [Decision Letter · Decision Letter 0]

27 Jan 2025

PONE-D-24-58988

Geography is a stronger predictor of diversification of monogenean parasites (Platyhelminthes) than host relatedness in characid fishes of Middle America

PLOS ONE

Dear Dr. Alda,

Thank you for submitting your manuscript to PLOS ONE. After careful consideration, we feel that it has merit but does not fully meet PLOS ONE’s publication criteria as it currently stands. Therefore, we invite you to submit a revised version of the manuscript that addresses the points raised during the review process.

As you will see, all three reviews were quite positive. However, two of the reviewers had comments or concerns. Please review and if you do not agree with any of these, please explain carefully to avoid delays in the review process.

We look forward to receiving your revised manuscript.

Kind regards,

Windsor E. Aguirre, Ph.D.

Academic Editor

PLOS ONE

3. Thank you for stating the following financial disclosure:  [Funding for sequencing was provided by the Smithsonian Institution’s DNA Barcode Network. E.F.M-F. received funding from a Smithsonian Tropical Research Institute postdoctoral fellowship. W.H.-R. was funded through a grant from the University of Tennessee at Chattanooga Center of Excellence in Applied Computational Science and Engineering (CEACSE) awarded to F.A.].  Please state what role the funders took in the study.  If the funders had no role, please state: "The funders had no role in study design, data collection and analysis, decision to publish, or preparation of the manuscript." If this statement is not correct you must amend it as needed.

4. Please note that your Data Availability Statement is currently missing [the DOI/accession number of each dataset]. If your manuscript is accepted for publication, you will be asked to provide these details on a very short timeline. We therefore suggest that you provide this information now, though we will not hold up the peer review process if you are unable.

Additional Editor Comments:

Thank you for submitting this paper to PLOS One. We look forward to your revisions.

Reviewers' comments:

Reviewer's Responses to Questions

**Comments to the Author**

1. Is the manuscript technically sound, and do the data support the conclusions?

Reviewer #1: Yes

Reviewer #2: Yes

Reviewer #3: Yes

2. Has the statistical analysis been performed appropriately and rigorously? 

Reviewer #1: No

Reviewer #2: Yes

Reviewer #3: Yes

3. Have the authors made all data underlying the findings in their manuscript fully available?

Reviewer #1: Yes

Reviewer #2: Yes

Reviewer #3: Yes

4. Is the manuscript presented in an intelligible fashion and written in standard English?

Reviewer #1: Yes

Reviewer #2: Yes

Reviewer #3: Yes

5. Review Comments to the Author

Reviewer #1: The manuscript numbered PONE-D-24-58988 and titled “Geography is a stronger predictor of diversification of monogenean parasites (Platyhelminthes) than host relatedness in characid fishes of Middle America” authored by Alda and colleagues. This study examines the relationship between characid fishes of the genus Astyanax and their monogenean parasites. The researchers constructed phylogeographic hypotheses for monogenean ectoparasites associated with four species of characid fishes from Panama, Nicaragua, and Mexico. They initially hypothesized that parasite diversity and community structure would correlate more strongly with host species than with geographic location. However, their findings concluded that geography or geographic location was a stronger predictor.

I was very pleased to read this paper, as it represents the first time this type of work has been done in the region. However, I have a number of major and minor concerns that the authors will need to address, and I hope these suggestions will help improve the manuscript.

Major concern:

In the Geographic analysis section of the methods, it is stated that PERMANOVA was used to test weather geographic location (Factor) or host identity (factor) better predicts parasite diversity and community structure (dependent variable). However, I believe that the way PERMANOVA test was set up, it is just telling if there are significant differences in parasite diversity (dependent variable) and community structure by geographic location (factor) or host identity (factor), so this needs to be fix.

Another important issue with the implementation of this PERMANOVAS is that no post-hoc test was applied to test for comparison between pairs of host or geographic locations. The result of the PERMANOVA was significant, but we don’t know if all comparisons are significant, remember that if a single comparison returns significant, we will have a significant p. value, but in reality, we don’t know how many comparisons are or are none significant.

The same comment applies to the second PERMANOVA in which authors try to elucidate if host species or geographic location predicted genetic similarity among individual parasites. I believe that here a simple mantel test would have given a better answer.

Minor concerns:

Line 101: Check journal citation style for (Choudhury et al. 2017). I believe it should be formatted as: “these biogeographical patterns [25]"

Line 103-4: "Lower Middle America (LMA)". This is more of a comment than a suggestion to change anything in the manuscript. My concern is that when one reads "Lower Middle America," it immediately evokes the southernmost part of Middle America, which suggests to me that the study area only includes Panama and Costa Rica. In many biological studies, the region encompassing Central America and southern Mexico is commonly referred to as Central America.

Line 121: The authors state that nine locations in Panama were sampled; however, the map in Figure 1a shows only seven locations.

Line 134: The reference to Matamoros et al. (2005) is not listed in the reference list. Instead, it is cited as: '28. Matamoros WA, McMahan CD, Chakrabarty P, Albert JS, Schaefer JF. Derivation of the freshwater fish fauna of Central America revisited: Myers’s hypothesis in the twenty-first century. Cladistics. 2015;31: 177–188.' Therefore, the correct citation should be: Matamoros et al. (2015).

Lines 134-7: The authors indicate in line 134 that this figure shows the results of a Maximum Likelihood analysis. However, in lines 136-137, they also mention providing Bayesian Posterior Probabilities, which suggests that this phylogeny is also the result of a Bayesian analysis. Please clarify this issue.

Table 1. The 'No. Parasite Species' column is not necessary, as the table is small enough for the reader to quickly discern the number of species per fish host. Additionally, the title 'Parasite Species' is causing some confusion as it is. I suggest centering this title for clarity.

Line 189: It is clear that the authors performed both Maximum Likelihood and Bayesian analyses; however, the way Figure 1 is presented suggests that only an ML analysis was conducted.

Table 2. Please explain the meaning of 'C', 'D', 'HS', 'L', and 'FD' in the table.

Line 247: “genetic distance matrix calculated in PAUP” authors need to specify which metric PAUP used in order to stimate distances¡

Line 274-5: Characithecium costaricensis (Price and Bussing 1967) Mendoza-Franco, Reina and Torchin 2009. I believe the authority for C. costaricensis is (Price and Bussing 1967), However, I do not understand what the role of Mendoza-Franco, Reina and Torchin 2009 citation is, please fix.

Line 296-7: This indicates that the ML and BI trees were not identical. However, earlier, you mentioned that the results were the same. Could you please clarify this discrepancy?

Lines 298-9: I suggest color coded the terminals in figure 2A the same way terminals were coded in figure 1B.

Line 368: The authors wrote: 'within Mexico'; however, I would recommend specifying 'southern Mexico.' Mexico is a large country, and the study's samples were collected from specific regions within the southern states. Using 'southern Mexico' would provide greater clarity and precision regarding the geographical scope of the study.

Line 406-409: Could the position of U. strombicirrus in these results be attributed to incomplete lineage sorting or any potential bias associated with relying solely on mitochondrial DNA data?"

Line 418-420: This section addresses the concerns I raised in my previous comment.

Other comments: The figure S1 is very helpful in understanding the results. I suggest inserting a smaller, color-coded map showing the areas of endemism. This addition would help readers geographically locate the tree terminals.

Reviewer #2: This is a high quality study using modern molecular tools to ask an interesting and well-defined question about host-parasite evolution, employing munificent and well-analyzed molecular data, with thorough scholarship and clearly-explained results. A model study illustrating the important conclusion that geographic location can be a stronger predictor of ecological association than the taxonomic identity of the host species.

Reviewer #3: The authors investigate and test factors driving the diversification of monogenean parasites in relation to their hosts, focusing on three species of the genus Astyanax and Eretmobrycon gonzalezoi. To achieve this, they analyzed mitochondrial genes—ATPase 6/8 from the fish and COI from several gill monogenean parasites—collected across multiple areas of endemism in Central America.

Using cophylogenetic analysis and modeling various scenarios involving the four host species and eleven geographic regions, they concluded that geography is a stronger predictor of parasite diversification than the genetic distance of the hosts. This finding challenges the traditional notion of parasite diversification through strict coevolution and expands our understanding of how parasites diversify in association with their hosts at a lower taxonomic level.

I find the article robust, and the conclusions are well-supported by the data. My primary comment concerns the use of the term “characid” throughout the text, which refers to an outdated classification of the family Characidae. I recommend revising this terminology to ensure clarity and avoid confusion. Additional minor edits are detailed below.

LINE 2: Characidae has been split recently and Astyanax has been reassigned to the family Acestrorhamphidae based on UCE. (Eschmeyer's Catalog of Fishes), but see Melo et. al. (2024) (https://doi.org/10.1093/zoolinnean/zlae101). May want to consider and use a different term. A potential alternative could be “Characin”, but I leave to the authors to decide what is more appropriate.

LINE 30: Update all instances of the term “Characids” as appropriate.

LINE 84-85: Update sentences to reflect latest classification of the genus Astyanax within Acestrorhamphidae: Acestrorhamphinae . Here could provide context highlighting old and recent classification.

LINE 101: Change in text citation format for consistency for “(Choudhury et al. 2017).”

LINE 120: Add parenthesis to “Eigenmann 1907” for consistency with other species authors.

LINE 121: Update “Characid”. Eretmobrycon gonzalezoi is now in Stevardiidae: Landoninae. See Melo et al. (2024).

LINE 134: Maybe you mean “Matamoros et al. 2015” instead of “Matamoros et al. 2005”. Add number of citation.

LINE 173: Typo “originally JB3 in 12/20/2024 6:16:00 AM”

FIG 1A: “A. ruberrimus (2)” appear in the figure but not mentioned in the text.

FIG 1A: For clarity could label the locations within the areas of endemism.

TABLE 2: Add the meaning of the event letters in the caption or text. E.g., Cospeciation (C), Duplication (D), etc…

LINE 242: Often Latin words such as “sensu” are italicized, however, may depend on the journal. Worth asking.

LINE 252: Add R version.

LINE 261: remove “characid”

LINE 303: Update family “characid”

LINE 311: Update family “characid”

LINE 364: Define the term “characids"

LINE 493: Typo. I think it should be “Fig 2B”

6. PLOS authors have the option to publish the peer review history of their article (what does this mean? ). If published, this will include your full peer review and any attached files.

**Do you want your identity to be public for this peer review?** For information about this choice, including consent withdrawal, please see our Privacy Policy .

Reviewer #1: No

Reviewer #2: No

Reviewer #3: No

---

## [Author Response · Author response to Decision Letter 1]

9 Mar 2025

The response to reviewers is also included in an attached file.

Reviewer #1:

The manuscript numbered PONE-D-24-58988 and titled “Geography is a stronger predictor of diversification of monogenean parasites (Platyhelminthes) than host relatedness in characid fishes of Middle America” authored by Alda and colleagues. This study examines the relationship between characid fishes of the genus Astyanax and their monogenean parasites. The researchers constructed phylogeographic hypotheses for monogenean ectoparasites associated with four species of characid fishes from Panama, Nicaragua, and Mexico. They initially hypothesized that parasite diversity and community structure would correlate more strongly with host species than with geographic location. However, their findings concluded that geography or geographic location was a stronger predictor.

I was very pleased to read this paper, as it represents the first time this type of work has been done in the region. However, I have a number of major and minor concerns that the authors will need to address, and I hope these suggestions will help improve the manuscript.

Major concern:

In the Geographic analysis section of the methods, it is stated that PERMANOVA was used to test weather geographic location (Factor) or host identity (factor) better predicts parasite diversity and community structure (dependent variable). However, I believe that the way PERMANOVA test was set up, it is just telling if there are significant differences in parasite diversity (dependent variable) and community structure by geographic location (factor) or host identity (factor), so this needs to be fix.

Another important issue with the implementation of this PERMANOVAS is that no post-hoc test was applied to test for comparison between pairs of host or geographic locations. The result of the PERMANOVA was significant, but we don’t know if all comparisons are significant, remember that if a single comparison returns significant, we will have a significant p. value, but in reality, we don’t know how many comparisons are or are none significant.

The same comment applies to the second PERMANOVA in which authors try to elucidate if host species or geographic location predicted genetic similarity among individual parasites. I believe that here a simple mantel test would have given a better answer.

>>We thank the reviewer for their insightful comments on the statistical analyses. Following their recommendations, we have reanalyzed the data by re-running the PERMANOVA for both community dissimilarity and parasite genetic distance.

1. We have revised the methods section to more accurately reflect the purpose of the analysis:

P4, L244–247 “To test whether parasite diversity and community structure are shaped by geographic location and/or host identity, we conducted a permutational multivariate analysis of variance (PERMANOVA) using the adonis2 function in the R package vegan v.2.6–4 [39].”

2. We ran independent models, including all explanatory variables together and separately, to compare their R² values and evaluate the relative effects of geography and host species (P4, L256-259).

3. To further disentangle these effects, we conducted a variance partitioning analysis to quantify the individual contribution of each explanatory variable while controlling for the others (P4, L259-261).

4. As suggested, we also performed post hoc pairwise PERMANOVA tests (pairwise.adonis) to compare parasite communities between pairs of host species and geographic locations (P5, L263-267).

5. Finally, we conducted Mantel tests to examine the relationships between community dissimilarities or genetic distances and geographic distances. However, we ultimately decided not to include these results in the final manuscript. Our interpretation was that the observed significance was primarily driven by the aggregation of identical sequences in certain regions, such as the Grijalva-Usumacinta. This pattern is evident in the figures below, where data points in the lower-left corner correspond to parasite samples from the same species with identical or highly similar sequences in this region. When these samples were excluded, the test became non-significant. Given this, we determined that logistic regression provided a more appropriate approach for assessing parasite species co-occurrence within hosts or regions.

Minor concerns:

Line 101: Check journal citation style for (Choudhury et al. 2017). I believe it should be formatted as: “these biogeographical patterns [25]"

>> We have fixed the reference formatting.

Line 103-4: "Lower Middle America (LMA)". This is more of a comment than a suggestion to change anything in the manuscript. My concern is that when one reads "Lower Middle America," it immediately evokes the southernmost part of Middle America, which suggests to me that the study area only includes Panama and Costa Rica. In many biological studies, the region encompassing Central America and southern Mexico is commonly referred to as Central America.

>> We appreciate the reviewer’s feedback and recognize the potential for confusion. Initially, we referred only to Lower Middle America (LMA) because the paper primarily focused on Panama. However, we acknowledge the importance of clarity and have revised the sentence to refer to Middle America (MA) more broadly, while retaining the term LMA only when specifically discussing the biogeographical regions within Panama (P5, L104-105).

Line 121: The authors state that nine locations in Panama were sampled; however, the map in Figure 1a shows only seven locations.

>> We have corrected this mistake. There are only eight (not nine) sampling locations in Panama. Additionally, two sampling points in Tuira appeared overlapped due to the map’s scale. We have corrected the text (P6, L122) and adjusted Figure 1 to ensure all sampling points are clearly visible.

Line 134: The reference to Matamoros et al. (2005) is not listed in the reference list. Instead, it is cited as: '28. Matamoros WA, McMahan CD, Chakrabarty P, Albert JS, Schaefer JF. Derivation of the freshwater fish fauna of Central America revisited: Myers’s hypothesis in the twenty-first century. Cladistics. 2015;31: 177–188.' Therefore, the correct citation should be: Matamoros et al. (2015).

>> P7, L136: We have corrected this. The appropriate reference is indeed Matamoros et al. (2015).

Lines 134-7: The authors indicate in line 134 that this figure shows the results of a Maximum Likelihood analysis. However, in lines 136-137, they also mention providing Bayesian Posterior Probabilities, which suggests that this phylogeny is also the result of a Bayesian analysis. Please clarify this issue.

>> P7, L136-139: We have clarified the phylogenetic tree results. Maximum Likelihood (ML) and Bayesian analyses were used to infer phylogenetic relationships based on mitochondrial DNA sequences from fish hosts and parasites. Both methods produced congruent topologies for relationships with moderate to high support. Given that discrepancies between inference methods occurred only at poorly supported nodes, we chose to present the ML topology, as these differences were not statistically robust and therefore not relevant to the discussion.

In Figures 1 and 2, we have indicated that the topology corresponds to the ML tree but support values are shown for both analyses. In the results section we have also indicated that the trees are overall congruent and we only show the ML topology.

Table 1. The 'No. Parasite Species' column is not necessary, as the table is small enough for the reader to quickly discern the number of species per fish host. Additionally, the title 'Parasite Species' is causing some confusion as it is. I suggest centering this title for clarity.

>> We have modified Table 1 as suggested by the reviewer.

Line 189: It is clear that the authors performed both Maximum Likelihood and Bayesian analyses; however, the way Figure 1 is presented suggests that only an ML analysis was conducted.

>> We have modified the caption of figure 1 (P7, L136-139) to indicate that only the topology of the ML is shown. Also, in the results we have underscored the overall congruence of phylogenetic analyses and therefore the support values are shown for both analyses over the ML topology.

Table 2. Please explain the meaning of 'C', 'D', 'HS', 'L', and 'FD' in the table.

>> We have added a description of the abbreviations used.

Line 247: “genetic distance matrix calculated in PAUP” authors need to specify which metric PAUP used in order to stimate distances

>> We have indicated the best nucleotide substitution models estimated by jModelTest and used to calculate the genetic distance matrices are: HKY+I for ATPase 8 and TIM1+G for ATPase 6 (P5, L281-282) and TIM3+I+G for COI (P6, L306).

Line 274-5: Characithecium costaricensis (Price and Bussing 1967) Mendoza-Franco, Reina and Torchin 2009. I believe the authority for C. costaricensis is (Price and Bussing 1967), However, I do not understand what the role of Mendoza-Franco, Reina and Torchin 2009 citation is, please fix.

>> We understand the reviewer’s comment given the taxonomic confusion in this group. Characithecium costaricensis was originally described as Cleidodiscus costaricensis by Price and Bussing (1967) and later transferred to the genus Urocleidoides by Kritsky and Leiby (1972) as Urocleidoides costaricensis. In (2009) Mendoza-Franco et al. 2009 erected Characithecium and accommodated this species as Characithecium costaricensis.

Since the species was originally described in another genus (i.e., Cleidodiscus) this is indicated by placing author and date in parenthesis (i.e. Price and Bussing 1967) followed by Mendoza-Franco, Reina and Torchin 2009, who made the last taxonomic change.

In this case, we have followed Article 51 from the ICZN for the Citation of names of authors:

51.1. Optional use of names of authors

Recommendation 51G. Citation of person making new combination. If it is desired to cite both the author of a species-group nominal taxon and the person who first transferred it to another genus, the name of the person forming the new combination should follow the parentheses that enclose the name of the author of the species-group name (and the date, if cited; see Recommendation 22A.3).

See also: https://www.aje.com/arc/editing-tip-scientific-names-species/

Line 296-7: This indicates that the ML and BI trees were not identical. However, earlier, you mentioned that the results were the same. Could you please clarify this discrepancy?

>> We have clarified that the ML and BI host and parasite trees are not identical. However, they are largely congruent in the major relationships recovered, with differences primarily occurring at unsupported nodes (P6-7, L306-308). By definition, nodes with low support are unresolved and could correspond to any topology; therefore, we do not interpret them as conflicting relationships. In the Discussion, we acknowledge the limitations of the molecular markers used for phylogenetic inference, as they were originally designed for species identification rather than resolving deep evolutionary relationships.

Lines 298-9: I suggest color coded the terminals in figure 2A the same way terminals were coded in figure 1B.

>> We have changed Figure 2A as suggested by the reviewer

Line 368: The authors wrote: 'within Mexico'; however, I would recommend specifying 'southern Mexico.' Mexico is a large country, and the study's samples were collected from specific regions within the southern states. Using 'southern Mexico' would provide greater clarity and precision regarding the geographical scope of the study.

>> We have corrected this and now the sentence reads: “In local communities within southern Mexico, C. costaricensis and P. heteroancistrium were dominant species [6].” (P11, L403)

Line 406-409: Could the position of U. strombicirrus in these results be attributed to incomplete lineage sorting or any potential bias associated with relying solely on mitochondrial DNA data?"

>> We agree with the reviewer’s comment. As noted in their subsequent comment and discussed in the manuscript, there are inherent limitations in the relationships inferred using the current molecular markers.

Line 418-420: This section addresses the concerns I raised in my previous comment.

>> See comment above.

Other comments: The figure S1 is very helpful in understanding the results. I suggest inserting a smaller, color-coded map showing the areas of endemism. This addition would help readers geographically locate the tree terminals.

>> We have added a map to figure S1 showing the areas of endemism.

Reviewer #2:

This is a high quality study using modern molecular tools to ask an interesting and well-defined question about host-parasite evolution, employing munificent and well-analyzed molecular data, with thorough scholarship and clearly-explained results. A model study illustrating the important conclusion that geographic location can be a stronger predictor of ecological association than the taxonomic identity of the host species.

Reviewer #3:

The authors investigate and test factors driving the diversification of monogenean parasites in relation to their hosts, focusing on three species of the genus Astyanax and Eretmobrycon gonzalezoi. To achieve this, they analyzed mitochondrial genes—ATPase 6/8 from the fish and COI from several gill monogenean parasites—collected across multiple areas of endemism in Central America.

Using cophylogenetic analysis and modeling various scenarios involving the four host species and eleven geographic regions, they concluded that geography is a stronger predictor of parasite diversification than the genetic distance of the hosts. This finding challenges the traditional notion of parasite diversification through strict coevolution and expands our understanding of how parasites diversify in association with their hosts at a lower taxonomic level.

I find the article robust, and the conclusions are well-supported by the data. My primary comment concerns the use of the term “characid” throughout the text, which refers to an outdated classification of the family Characidae. I recommend revising this terminology to ensure clarity and avoid confusion. Additional minor edits are detailed below.

LINE 2: Characidae has been split recently and Astyanax has been reassigned to the family Acestrorhamphidae based on UCE. (Eschmeyer's Catalog of Fishes), but see Melo et. al. (2024) (https://doi.org/10.1093/zoolinnean/zlae101). May want to consider and use a different term. A potential alternative could be “Characin”, but I leave to the authors to decide what is more appropriate.

>> We thank the reviewer for this comment. We had used “characid” as a common name often used to refer to members of the former family Characidae, as in Melo et al. 2024. However, we have followed their recommendation and changed this term for “characin” throughout the manuscript. Also, we have corrected the names of the families where Astyanax and Eretmobrycon are currently assigned (see comments below).

P1 L1-2: Geography is a stronger predictor of diversification of monogenean parasites (Platyhelminthes) than host relatedness in characin fishes of Middle America

P4 L84-86: In this study, we examine characin fish species (Acestrorhamphidae: Acestrorhamphinae and Stevardiidae: Landoninae sensu Melo et al. [18]) and their monogenean gill parasites (Platyhelminthes: Monogenea).

Bruno F Melo, Rafaela P Ota, Ricardo C Benine, Fernando R Carvalho, Flavio C T Lima, George M T Mattox, Camila S Souza, Tiago C Faria, Lais Reia, Fabio F Roxo, Martha Valdez-Moreno, Thomas J Near, Claudio Oliveira, Phylogenomics of Characidae, a hyper-diverse Neotropical freshwater fish lineage, with a phylogenetic classification including four families (Teleostei: Characiformes), Zoological Journal of the Linnean Society, Volume 202, Issue 1, September 2024, zlae101, https://doi.org/10.1093/zoolinnean/zlae101

LINE 30: Update all instances of the term “Characids” as appropriate.

>> As suggested, we have changed all the instances of the term “characid” to “characin”.

LINE 84-85: Update sentences to reflect latest classification of the genus Astyanax within Acestrorhamphi

---

## [Editor Report · Decision Letter 1]

11 Mar 2025

Geography is a stronger predictor of diversification of monogenean parasites (Platyhelminthes) than host relatedness in characin fishes of Middle America

PONE-D-24-58988R1

Dear Dr. Alda,

We’re pleased to inform you that your manuscript has been judged scientifically suitable for publication and will be formally accepted for publication once it meets all outstanding technical requirements.

Kind regards,

Windsor E. Aguirre, Ph.D.

Academic Editor

PLOS ONE
---

## [Editor Report · Acceptance letter]

PONE-D-24-58988R1

PLOS ONE

Dear Dr. Alda,

I'm pleased to inform you that your manuscript has been deemed suitable for publication in PLOS ONE. Congratulations! Your manuscript is now being handed over to our production team.

Kind regards,

on behalf of

Dr. Windsor E. Aguirre

Academic Editor

PLOS ONE